# COMBINATORIAL PURE EXPLORATION OF CAUSAL BANDITS

**Nuoya Xiong**
Institute for Interdisciplinary Information Sciences, Tsinghua University
Beijing, China
`xiongny20@mails.tsinghua.edu.cn`

**Wei Chen**
Microsoft Research
Beijing, China
`weic@microsoft.com`

## ABSTRACT

The combinatorial pure exploration of causal bandits is the following online learning task: given a causal graph with unknown causal inference distributions, in each round we choose a subset of variables to intervene or do no intervention, and observe the random outcomes of all random variables, with the goal that using as few rounds as possible, we can output an intervention that gives the best (or almost best) expected outcome on the reward variable $Y$ with probability at least $1 - \delta$, where $\delta$ is a given confidence level. We provide the first gap-dependent and fully adaptive pure exploration algorithms on two types of causal models — the binary generalized linear model (BGLM) and general graphs. For BGLM, our algorithm is the first to be designed specifically for this setting and achieves polynomial sample complexity, while all existing algorithms for general graphs have either sample complexity exponential to the graph size or some unreasonable assumptions. For general graphs, our algorithm provides a significant improvement on sample complexity, and it nearly matches the lower bound we prove. Our algorithms achieve such improvement by a novel integration of prior causal bandit algorithms and prior adaptive pure exploration algorithms, the former of which utilize the rich observational feedback in causal bandits but are not adaptive to reward gaps, while the latter of which have the issue in reverse.

## 1 INTRODUCTION

Stochastic multi-armed bandits (MAB) is a classical framework in sequential decision making (Robbins, 1952). In each round, a learner selects one arm based on the reward feedback from the previous rounds, and receives a random reward of the selected arm sampled from an unknown distribution, with the goal of accumulating as much rewards as possible. Pure exploration is an important variant of the multi-armed bandit problem, where the goal is not to accumulate reward but to identify the best arm through possibly adaptive explorations of arms.

Causal bandits, first introduced by Lattimore et al. (2016), integrates causal inference (Pearl, 2009) with multi-armed bandits. In causal bandits, we have a causal graph structure $G = (\boldsymbol{X} \cup \{Y\} \cup \boldsymbol{U}, E)$, where $\boldsymbol{X} \cup \{Y\}$ are observable causal variables with $Y$ being a special reward variable, $\boldsymbol{U}$ are unobserved hidden variables, and $E$ is the set of causal edges between pairs of variables. For simplicity, we consider binary variables in this paper. The arms are the interventions on variables $\boldsymbol{S} \subseteq \boldsymbol{X}$ together with the choice of null intervention (natural observation), i.e. the action set is $A \subseteq \{a = do(\boldsymbol{S} = \boldsymbol{s}) \mid \boldsymbol{S} \subseteq \boldsymbol{X}, \boldsymbol{s} \in \{0, 1\}^{|\boldsymbol{S}|}\}$ with $do() \in A$, where $do(\boldsymbol{S} = \boldsymbol{s})$ is the standard notation for intervening the causal graph by setting $\boldsymbol{S}$ to $\boldsymbol{s}$ (Pearl, 2009), and $do()$ means null intervention. The reward of an action $a$ is the random outcome of $Y$, and thus the expected reward is $\mathbb{E}[Y \mid a = do(\boldsymbol{S} = \boldsymbol{s})]$. In each round, one action in $A$ is played, and the random outcomes of all variables in $\boldsymbol{X} \cup \{Y\}$ are observed. Given the causal graph $G$ but without knowing the distributions among nodes, the task of combinatorial pure exploration (CPE) of causal bandits is to select actions

in each round, observe the feedback from all observable random variables, so that in the end the learner can identify the best or nearly best actions. Causal bandits are useful in many real scenarios. In drug testing, the physicians wants to adjust the dosage of some particular drugs to treat the patient. In policy design, the policy-makers select different actions to reduce the spread of disease.

Existing studies on CPE of causal bandits either requires the knowledge of $P(\boldsymbol{Pa}(Y) \mid a)$ for all action $a$ or only consider causal graphs without hidden variables, and the algorithms proposed are not fully adaptive to reward gaps (Lattimore et al., 2016; Yabe et al., 2018). In this paper, we study fully adaptive pure exploration algorithms and analyze their gap-dependent sample complexity bounds in the fixed-confidence setting. More specifically, given a confidence bound $\delta \in (0, 1)$ and an error bound $\varepsilon$, we aim at designing adaptive algorithms that output an action such that with probability at least $1 - \delta$, the expected reward difference between the output and the optimal action is at most $\varepsilon$. The algorithms should be fully adaptive in the follow two senses. First, it should adapt to the reward gaps between suboptimal and optimal actions similar to existing adaptive pure exploration bandit algorithms, such that actions with larger gaps should be explored less. Second, it should adapt to the observational data from causal bandit feedback, such that actions with enough observations already do not need further interventional rounds for exploration, similar to existing causal bandit algorithms. We are able to integrate both types of adaptivity into one algorithmic framework, and with interaction between the two aspects, we achieve better adaptivity than either of them alone.

First we introduce a particular term named gap-dependent observation threshold, which is a non-trivial gap-dependent extension for a similar term in Lattimore et al. (2016). Then we provide two algorithms, one for the binary generalized linear model (BGLM) and one for the general model with hidden variables. The sample complexity of both algorithms contains the gap-dependent observation threshold that we introduced, which shows significant improvement comparing to the prior work. In particular, our algorithm for BGLM achieves a sample complexity polynomial to the graph size, while all prior algorithms for general graphs have exponential sample complexity; and our algorithm for general graphs match a lower bound we prove in the paper. To our best knowledge, our paper is the first work considering a CPE algorithm specifically designed for BGLM, and the first work considering CPE on graphs with hidden variables, while all prior studies either assume no hidden variables or assume knowing distribution $P(\boldsymbol{Pa}(Y) \mid a)$ for the parents of reward variable $\boldsymbol{Pa}(Y)$ and all action $a$, which is not a reasonable assumption.

To summarize, our contribution is to propose the first set of CPE algorithms on causal graphs with hidden variables and fully adaptive to both the reward gaps and the observational causal data. The algorithm on BGLM is the first to achieve a gap-dependent sample complexity polynomial to the graph size, while the algorithm for general graphs improves significantly on sample complexity and matches a lower bound. Due to the space constraint, further materials including experimental results, an algorithm for the fixed-budget setting, and all proofs are moved to the appendix.

**Related Work.**  Causal bandit is proposed by Lattimore et al. (2016), who discuss the simple regret for parallel graphs and general graphs with known probability distributions $P(\boldsymbol{Pa}(Y) \mid a)$. Nair et al. (2021) extend algorithms on parallel graphs to graphs without back-door paths, and Maiti et al. (2021) extend the results to the general graphs. All of them either regard $P(\boldsymbol{Pa}(Y) \mid a)$ as prior knowledge, or consider only atomic intervention. The study by Yabe et al. (2018) is the only one considering the general graphs with combinatorial action set, but their algorithm cannot work on causal graphs with hidden variables. All the above pure exploration studies consider simple regret criteria that is not gap-dependent. Cumulative regret is considered in (Lu et al., 2020; Nair et al., 2021; Maiti et al., 2021). To our best knowledge, Sen et al. (2017) is the only one discussing gap-dependent bound for pure exploration of causal bandits for the fixed-budget setting, but it only considers the soft interventions (changing conditional distribution $P(X|\boldsymbol{Pa}(X))$) on *one single node*, which is different from causal bandits defined in Lattimore et al. (2016).

Pure exploration of multi-armed bandit has been extensively studied in the fixed-confidence or fixed-budget setting (Audibert et al., 2010; Kalyanakrishnan et al., 2012; Jamieson et al., 2013; Jamieson & Nowak, 2014). PAC pure exploration is a generalized setting aiming to find the $\varepsilon$-optimal arm instead of exactly optimal arm (Even-Dar et al., 2002; Mannor & Tsitsiklis, 2004). In this paper, we utilize the adaptive LUCB algorithm in (Kalyanakrishnan et al., 2012). CPE has also been studied for multi-armed bandits and linear bandits, etc.(Karnin et al. (2013b); Chen et al. (2014); Du et al. (2021)), but the feedback model in these studies either have feedback at the base arm level or have full or partial bandit feedback, which are all different from the causal bandit feedback.

The binary generalized linear model (BGLM) is studied in (Li et al., 2017; Feng & Chen, 2022) for cumulative regret MAB problems. We borrow the maximum likelihood estimation method and its result in (Li et al., 2017; Feng & Chen, 2022) for our BGLM part, but its integration with our adaptive sampling algorithm for the pure exploration setting is new.

## 2 MODEL AND PRELIMINARIES

**Causal Models.** From Pearl (2009), a causal graph $G = (\boldsymbol{X} \cup \{Y\} \cup \boldsymbol{U}, E)$ is a directed acyclic graph (DAG) with a set of observed random variables $\boldsymbol{X} \cup \{Y\}$ and a set of hidden random variables $\boldsymbol{U}$, where $\boldsymbol{X} = \{X_1, \cdots, X_n\}$, $\boldsymbol{U} = \{U_1, \cdots, U_k\}$ are two set of variables and $Y$ is the special reward variable without outgoing edges. In this paper, for simplicity, we only consider that $X_i$'s and $Y$ are binary random variables with support $\{0, 1\}$. For any node $V$ in $G$, we denote the set of its parents in $G$ as $\boldsymbol{Pa}(V)$. The set of values for $\boldsymbol{Pa}(X)$ is denoted by $\boldsymbol{pa}(X)$. The causal influence is represented by $P(V \mid \boldsymbol{Pa}(V))$, modeling the fact that the probability distribution of a node $V$'s value is determined by the value of its parents. Henceforth, when we refer to a causal graph, we mean both its graph structure $(\boldsymbol{X} \cup \{Y\} \cup \boldsymbol{U}, E)$ and its causal inference distributions $P(V \mid \boldsymbol{Pa}(V))$ for all $V \in \boldsymbol{X} \cup \{Y\} \cup \boldsymbol{U}$. A parallel graph $G = (\boldsymbol{X} \cup \{Y\}, E)$ is a special class of causal graphs with $\boldsymbol{X} = \{X_1, \cdots, X_n\}$, $\boldsymbol{U} = \emptyset$ and $E = \{X_1 \to Y, X_2 \to Y, \cdots, X_n \to Y\}$. An *intervention* $do(\boldsymbol{S} = \boldsymbol{s})$ in the causal graph $G$ means that we set the values of a set of nodes $\boldsymbol{S} \subseteq \boldsymbol{X}$ to $\boldsymbol{s}$, while other nodes still follow the $P(V \mid \boldsymbol{Pa}(V))$ distributions. An *atomic intervention* means that $|\boldsymbol{S}| = 1$. When $\boldsymbol{S} = \emptyset$, $do(\boldsymbol{S} = \boldsymbol{s})$ is the null intervention denoted as $do()$, which means we do not set any node to any value and just observe all nodes' values.

In this paper, we also study a parameterized model with no hidden variables: *the binary generalized linear model* (BGLM). Specifically, in BGLM, we have $\boldsymbol{U} = \emptyset$, and $P(X = 1 \mid \boldsymbol{Pa}(X) = \boldsymbol{pa}(X)) = f_X(\boldsymbol{\theta}_X \cdot \boldsymbol{pa}(X)) + e_X$, where $f_X$ is a strictly increasing function, $\boldsymbol{\theta}_X \in \mathbb{R}^{\boldsymbol{Pa}(X)}$ is the unknown parameter vector for $X$, $e_X$ is a zero-mean bounded noise variable that guarantees the resulting probability to be within $[0, 1]$. To represent the intrinsic randomness of node $X$ not caused by its parents, we denote $X_1 = 1$ as a global variable, which is a parent of all nodes.

**Combinatorial Pure Exploration of Causal Bandits.** Combinatorial pure exploration (CPE) of causal bandits describes the following setting and the online learning task. The causal graph structure is known but the distributions $P(V|\boldsymbol{Pa}(V))$'s are unknown. The action (arm) space $\boldsymbol{A}$ is a subset of possible interventions on combinatorial sets of variables, plus the null intervention, that is, $\boldsymbol{A} \subseteq \{do(\boldsymbol{S} = \boldsymbol{s}) \mid \boldsymbol{S} \subseteq \boldsymbol{X}, \boldsymbol{s} \in \{0, 1\}^{|\boldsymbol{S}|}\}$ and $\{do()\} \in \boldsymbol{A}$. For action $a = do(\boldsymbol{S} = \boldsymbol{s})$, define $\mu_a = \mathbb{E}[Y \mid do(\boldsymbol{S} = \boldsymbol{s})]$ to be the expected reward of action $do(\boldsymbol{S} = \boldsymbol{s})$. Let $\mu^* = \max_{a \in \boldsymbol{A}} \mu_a$.

In each round $t$, the learning agent plays one action $a \in \boldsymbol{A}$, observes the sample values $\boldsymbol{X}_t = (X_{t,1}, X_{t,2} \cdots, X_{t,n})$ and $Y_t$ of all observed variables. The goal of the agent is to interact with the causal model with as small number of rounds as possible to find an action with the maximum expected reward $\mu^*$. More precisely, we mainly focus on the following PAC pure exploration with the gap-dependent bound in the *fixed-confidence setting*. In this setting, we are given a confidence parameter $\delta \in (0, 1)$ and an error parameter $\varepsilon \in [0, 1)$, and we want to adaptively play actions over rounds based on past observations, terminate at a certain round and output an action $a^o$ to guarantee that $\mu^* - \mu_{a^o} \leq \varepsilon$ with probability at least $1 - \delta$. The metric for this setting is sample complexity, which is the number of rounds needed to output a proper action $a^o$. Note that when $\varepsilon = 0$, the PAC setting is reduced to the classical pure exploration setting. We also consider the *fixed budget setting* in the appendix, where given an exploration round budget $T$ and an error parameter $\varepsilon \in [0, 1)$, the agent is trying to adaptively play actions and output an action $a^o$ at the end of round $T$, so that the error probability $\Pr\{\mu_{a^o} < \mu^* - \varepsilon\}$ is as small as possible.

We study the gap-dependent bounds, meaning that the performance measure is related to the reward gap between the optimal and suboptimal actions, as defined below. Let $a^*$ be one of the optimal arms. For each arm $a$, we define the gap of $a$ as

$$\Delta_a = \begin{cases} \mu_{a^*} - \max_{a \in \boldsymbol{A} \setminus \{a^*\}} \{\mu_a\}, & a = a^*; \\ \mu_{a^*} - \mu_a, & a \neq a^*. \end{cases} \tag{1}$$

We further sort the gaps $\Delta_a$'s for all arms and assume $\Delta^{(1)} \leq \Delta^{(2)} \cdots \leq \Delta^{(|\boldsymbol{A}|)}$, where $\Delta^{(1)}$ is also denoted as $\Delta_{\min}$.

## 3 GAP-DEPENDENT OBSERVATION THRESHOLD

In this section, we introduce the key concept of gap-dependent observation threshold, which is instrumental to the fix-confidence algorithms in the next two sections. Intuitively, it describes for any action $a$ whether we can derive its reward from pure observations of the causal model.

We assume that $X_i$'s are binary random variables. First, we describe terms $q_a \in [0, 1]$ for each action $a \in \boldsymbol{A}$, which can have different definitions in different settings. Intuitively, $q_a$ represents how easily the action $a$ is to be estimated by observation. For example, in Lattimore et al. (2016), for parallel graph with action set $\boldsymbol{A} = \{do(X_i = x) \mid 1 \le i \le n, x \in \{0, 1\}\} \cup \{do()\}$, for action $a = do(X_i = x)$, $q_a = P(X_i = x)$ represents the probability for action $do(X_i = x)$ to be observed, since in parallel graph we have $P(Y \mid X_i = x) = P(Y \mid do(X_i = x))$. Thus, when $q_a = P(X_i = x)$ is larger, it is easier to estimate $P(Y \mid do(X_i = x))$ by observation. We will instantiate $q_a$'s for BGLM and general graphs in later sections. For $a = do()$, we always set $q_a = 1$. Then, for set $q_a, a \in \boldsymbol{A}$ we define the *observation thershold* as follows:

**Definition 1** (Observation threshold Lattimore et al. (2016)). *For a given causal graph $G$ and its associated $\{q_a \mid a \in \boldsymbol{A}\}$, the* observation threshold $m$ *is defined as:*

$$m = \min\{\tau \in [|\boldsymbol{A}|] : |\{a \in \boldsymbol{A} \mid q_a < 1/\tau\}| \le \tau\}. \tag{2}$$

The observation threshold can be equivalently defined as follows: When we sort $\{q_a \mid a \in \boldsymbol{A}\}$ as $q^{(1)} \le q^{(2)} \le \cdots \le q^{|\boldsymbol{A}|}$, $m = \min\{\tau : q^{(\tau+1)} \ge \frac{1}{\tau}\}$. Note that $m \le |\boldsymbol{A}|$ always holds since $q_{do()} = 1$. In some cases, $m \ll |\boldsymbol{A}|$. For example, in parallel graph, when $P(X_i = 1) = P(X_i = 0) = \frac{1}{2}$ for all $i \in [n]$, $q_{do(X_i=1)} = q_{do(X_i=0)} = \frac{1}{2}$, $q_{do()} = 1$. Then $m = 2 \ll 2n + 1 = |\boldsymbol{A}|$. Intuitively, when we collect passive observation data without intervention, arms corresponding to $q^{(j)}$ with $j \le m$ are under observed while arms corresponding to $q^{(j)}$ with $j > m$ are sufficiently observed and can be estimated accurately. Thus, for convenience we name $m$ as the observation threshold (the term is not given a name in Lattimore et al. (2016)).

In this paper, we improve the definition of $m$ to make it gap-dependent, which would lead to a better adaptive pure exploration algorithm and sample complexity bound. Before introducing the definition, we first define the term $H_r$. Sort the arm set as $q_{a_1} \cdot \max\{\Delta_{a_1}, \varepsilon/2\}^2 \le q_{a_2} \cdot \max\{\Delta_{a_2}, \varepsilon/2\}^2 \le \cdots \le q_{a_{|\boldsymbol{A}|}} \cdot \max\{\Delta_{a_{|\boldsymbol{A}|}}, \varepsilon/2\}^2$, then $H_r$ is defined by

$$H_r = \sum_{i=1}^{r} \frac{1}{\max\{\Delta_{a_i}, \varepsilon/2\}^2}. \tag{3}$$

**Definition 2** (Gap-dependent observation threshold). *For a given causal graph $G$ and its associated $q_a$'s and $\Delta_a$'s, the* gap-dependent observation threshold $m_{\varepsilon, \Delta}$ *is defined as:*

$$m_{\varepsilon, \Delta} = \min\left\{\tau \in [|\boldsymbol{A}|] : \left|\left\{a \in \boldsymbol{A} \middle| q_a \cdot \max\{\Delta_a, \varepsilon/2\}^2 < \frac{1}{H_\tau}\right\}\right| \le \tau\right\}. \tag{4}$$

The Gap-dependent observation threshold can be regarded as a generalization of the observation threshold. Intuitively, when considering the gaps, $q_a \cdot \max\{\Delta_a, \varepsilon/2\}^2$ represents how easily the action $a$ would to be distinguished from the optimal arm. To show the relationship between $m_{\varepsilon, \Delta}$ and $m$, we provide the following lemma. The proof of Lemma is in the supplementary material.

**Lemma 1.** $m_{\varepsilon, \Delta} \le 2m$.

Lemma 1 shows that $m_{\varepsilon, \Delta} = O(m)$. In many real scenarios, $m_{\varepsilon, \Delta}$ might be much smaller than $m$. Consider some integer $n$ with $4 < n < |\boldsymbol{A}|$, $\epsilon < 1/n$, $q_a = \frac{1}{n}$ for $a \in \boldsymbol{A} \setminus \{do()\}$ and $q_{do()} = 1$. Then $m = n$. Now we consider $\Delta_{a_1} = \Delta_{a_2} = \frac{1}{n}$, while other arms $a$ have $\Delta_a = \frac{1}{2}$. Then $H_r \ge n^2$ for all $r \ge 1$. Then for $a \ne a_1, a_2$, we have $q_a \cdot \max\{\Delta_a, \varepsilon/2\}^2 \ge \frac{1}{4n} > \frac{1}{H_r}$, which implies that $m_{\varepsilon, \Delta} = 2$. This lemma will be used to show that our result improves previous causal bandit algorithm in Lattimore et al. (2016).

## 4 COMBINATORIAL PURE EXPLORATION FOR BGLM

In this section, we discuss the pure exploration for BGLM, a general class of causal graphs with a linear number of parameters, as defined in Section 2. In this section, we assume $\boldsymbol{U} = \emptyset$. Let

$\boldsymbol{\theta}^* = (\boldsymbol{\theta}_X^*)_{X \in \boldsymbol{X} \cup \{Y\}}$ be the vector of all weights. Since $X_1$ is a global variable, we only need to consider the action set $\boldsymbol{A} \subseteq \{do(\boldsymbol{S} = \boldsymbol{s}) \mid \boldsymbol{S} \subseteq \boldsymbol{X} \setminus \{X_1\}, \boldsymbol{s} \in \{0,1\}^{|\boldsymbol{S}|}\}$. Following Li et al. (2017); Feng & Chen (2022), we have three assumptions:

**Assumption 1.** *For any $X \in \boldsymbol{X} \cup \{Y\}$, $f_X$ is twice differentiable. Its first and second order derivatives can be upper bounded by constant $M^{(1)}$ and $M^{(2)}$.*

**Assumption 2.** $\kappa := \inf_{X \in \boldsymbol{X} \cup \{Y\}, \boldsymbol{v} \in [0,1]^{Pa(X)}, ||\boldsymbol{\theta} - \boldsymbol{\theta}_X^*|| \leq 1} \dot{f}_X(\boldsymbol{v} \cdot \boldsymbol{\theta}) > 0$ *is a positive constant.*

**Assumption 3.** *There exists a constant $\eta > 0$ such that for any $X \in \boldsymbol{X} \cup \{Y\}$ and $X' \in \boldsymbol{Pa}(X)$, for any $\boldsymbol{v} \in \{0,1\}^{|\boldsymbol{Pa}(X)-2|}$ and $x \in \{0,1\}$, we have*

$$Pr[X' = x \mid \boldsymbol{Pa}(X) \setminus \{X', X_1\} = \boldsymbol{v}] \geq \eta. \tag{5}$$

Assumptions 1 and 2 are the classical assumptions in generalized linear model Li et al. (2017). Assumption 3 makes sure that each parent node of $X$ has some freedom to become 0 and 1 with a non-zero probability, even when the values of all other parents of $X$ are fixed, and it is originally given in Feng & Chen (2022) with additional justifications. Henceforth, we use $\sigma(\boldsymbol{\theta}, a)$ to denote the reward $\mu_a$ on parameter $\boldsymbol{\theta}$.

Our main algorithm, Causal Combinatorial Pure Exploration-BGLM (CCPE-BGLM), is given in Algorithm 1. The algorithm follows the LUCB framework Kalyanakrishnan et al. (2012), but has several innovations. In each round $t$, we play three actions and thus it corresponds to three rounds in the general CPE model. In each round $t$, we maintain $\hat{\mu}_{O,a}^t$ and $\hat{\mu}_{I,a}^t$ as the estimates of $\mu_a$ from the observational data and the interventional data, respectively. For each estimate, we maintain its confidence interval, $[L_{O,a}^t, U_{O,a}^t]$ and $[L_{I,a}^t, U_{I,a}^t]$ respectively.

At the beginning of round $t$, similar to LUCB, we find two candidate actions, one with the highest empirical mean so far, $a_h^{t-1}$; and one with the highest UCB among the rest, $a_l^{t-1}$. If the LCB of $a_h^{t-1}$ is higher than the UCB of $a_l^{t-1}$ with an $\varepsilon$ error, then the algorithm could stop and return $a_h^{t-1}$ as the best action. However, the second stopping condition in line 5 is new, and it is used to guarantee that the observational estimates $\hat{\mu}_{O,a}^t$'s are from enough samples. If the stopping condition is not met, we will do three steps. The first step is the novel observation step comparing to LUCB. In this step, we do the null intervention $do()$, collect observational data, use maximum-likelihood estimate adapted from Li et al. (2017); Feng & Chen (2022) to obtain parameter estimate $\hat{\theta}_t$, and then use $\hat{\theta}_t$ to compute observational estimate $\hat{\mu}_{O,a}^t = \sigma(\hat{\theta}_t, a)$ for all action $a$, where $\sigma(\hat{\theta}_t, a)$ means the reward for action $a$ on parameter $\hat{\theta}_t$. This can be efficiently done by following the topological order of nodes in $G$ and parameter $\hat{\theta}_t$. From $\hat{\mu}_{O,a}^t$, we obtain the confidence interval $[L_{O,a}^t, U_{O,a}^t]$ using the bonus term defined later in Eq.(8). In the second step, we play the two candidate actions $a_h^{t-1}$ and $a_l^{t-1}$ and update their interventional estimates and confidence intervals, as in LUCB. In the third step, we merge the two estimates together and set the final estimate $\hat{\mu}_a^t$ to be the mid point of the intersection of two confidence intervals. While the second step follows the LUCB, the first and the third step are new, and they are crucial for utilizing the observational data to obtain quick estimates for many actions at once.

Utilizing observational data has been explored in past causal bandit studies, but they separate the exploration from observations and the interventions into two stages (Lattimore et al., 2016; Nair et al., 2021), and thus their algorithms are not adaptive and cannot provide gap-dependent sample complexity bounds. Our algorithm innovation is in that we interleave the observation step and the intervention step naturally into the adaptive LUCB framework, so that we can achieve an adaptive balance between observation and intervention, achieving the best of both worlds.

To get the confidence bound for BGLM, we use the following lemma from Feng & Chen (2022):

**Lemma 2.** *For an action $a = do(\boldsymbol{S} = \boldsymbol{s})$ and any two weight vectors $\boldsymbol{\theta}$ and $\boldsymbol{\theta}'$, we have*

$$|\sigma(\boldsymbol{\theta}, a) - \sigma(\boldsymbol{\theta}', a)| \leq \mathbb{E}_{\boldsymbol{e}} \left[ \sum_{X \in N_{\boldsymbol{S}, Y}} |\boldsymbol{V}_X^\top (\boldsymbol{\theta}_X - \boldsymbol{\theta}'_X)| M^{(1)} \right], \tag{6}$$

*where $N_{\boldsymbol{S}, Y}$ is the set of all nodes that lie on all possible paths from $X_1$ to $Y$ excluding $\boldsymbol{S}$, $\boldsymbol{V}_X$ is the value vector of a sample of the parents of $X$ according to parameter $\boldsymbol{\theta}$, $M^{(1)}$ is defined in Assumption 1, and the expectation is taken on the randomness of the noise term $\boldsymbol{e} = (e_X)_{X \in \boldsymbol{X} \cup \{Y\}}$ of causal model under parameter $\boldsymbol{\theta}$.*

---

**Algorithm 1** CCPE-BGLM$(G, \boldsymbol{A}, \varepsilon, \delta, M^{(1)}, M^{(2)}, \kappa, \eta, c)$

---

1: Input:causal general graph $G$, action set $\boldsymbol{A}$, parameter $\varepsilon, \delta, M^{(1)}, M^{(2)}, \kappa, \eta, c$ in Assumptions 1,2, 3 and in Lemma 4 in supplementary material.
2: Initialize $M_{0,X} = I$ for all node $X$. $N_a = 0, \hat{\mu}_a^0 = 0, L_a^0 = -\infty, U_a^0 = \infty$ for arms $a \in \boldsymbol{A}$.
3: **for** $t = 1, 2, \cdots$, **do**
4:     $a_h^{t-1} = \text{argmax}_{a \in \boldsymbol{A}} \hat{\mu}_a^{t-1}, a_l^{t-1} = \text{argmax}_{a \in \boldsymbol{A} \backslash \{a_h^{t-1}\}} U_a^{t-1}$.
5:     **if** $U_{a_l^{t-1}}^{t-1} \leq L_{a_h^{t-1}}^{t-1} + \varepsilon$ and $t \geq \max\{\frac{cD}{\eta^2} \log \frac{nt^2}{\delta}, \frac{1024(M^{(2)})^2(4D^2-3)D}{\kappa^4\eta}(D^2 + \log \frac{3nt^2}{\delta})\}$ **then**
6:         **return** $a_h^{t-1}$.
7:     **end if**
8:     /* *Step 1. Conduct a passive observation and estimate from the observational data* */
9:     Perform action $do()$ and observe $\boldsymbol{X}_t$ and $Y_t$. For $a = do()$, $N_a = N_a + 1$.
10:     $\hat{\theta}_t = \text{BGLM-estimate}((\boldsymbol{X}_1, Y_1), \cdots, (\boldsymbol{X}_t, Y_t))$.
11:     For $a = do(\boldsymbol{S} = \boldsymbol{s}) \in \boldsymbol{A}$, calculate $\hat{\mu}_{O,a} = \sigma(\hat{\theta}_t, \boldsymbol{S})$, and $[L_{O,a}^t, U_{O,a}^t] = [\hat{\mu}_{O,a} - \beta_O^a(t), \hat{\mu}_{O,a} + \beta_O^a(t)]$. /* $\beta_O^a(t)$ is defined in Eq.(8) */
12:     /* *Step 2. Do two interventions and estimate from the interventional data* */
13:     Perform actions $a_l^{t-1}$ and $a_h^{t-1}$, get the reward $Y_t^{(l)}$ and $Y_t^{(h)}$.
14:     $N_{a_l^{t-1}} = N_{a_l^{t-1}} + 1, N_{a_h^{t-1}} = N_{a_h^{t-1}} + 1$.
15:     For $a \in \{a_l^{t-1}, a_h^{t-1}, do()\}$, update the empirical mean
16:     $\hat{\mu}_{I,a} = \sum_{j=1}^t \frac{1}{N_a}(\mathbb{I}\{a = a_l^{j-1}\}Y_j^{(l)} + \mathbb{I}\{a = a_h^{j-1}\}Y_j^{(h)} + \mathbb{I}\{a = do()\}Y_j)$ and $[L_{I,a}^t, U_{I,a}^t] = [\hat{\mu}_{I,a} - \beta_I(N_a), \hat{\mu}_{I,a} + \beta_I(N_a)]$. /* $\beta_I(t)$ is defined in Eq.(8) */
17:     /* *Step 3. Merge the observational estimate and the interventional estimate* */
18:     For $a \in \boldsymbol{A}$, calculate $[L_a^t, U_a^t] = [L_{O,a}^t, U_{O,a}^t] \cap [L_{I,a}^t, U_{I,a}^t]$ and $\hat{\mu}_a^t = \frac{L_a^t + U_a^t}{2}$.
19: **end for**

---

The key idea in the design and analysis of the algorithm is to divide the actions into two sets — the easy actions and the hard actions. Intuitively, the easy actions are the ones that can be easily estimated by observational data, while the hard actions require direction playing these actions to get accurate estimates. The quantity $q_a$ mentioned in Section 3 indicates how easy is action $a$, and it determines the gap-dependent observational threshold $m_{\varepsilon,\Delta}$ (Definition 2), which essentially gives the number of hard actions. In fact, the set of actions in Eq.(4) with $\tau = m_{\varepsilon,\Delta}$ is the set of hard actions and the rest are easy actions. We need to define $q_a$ representing the hardness of estimation for each $a$.

---

**Algorithm 2** BGLM-estimate

---

1: Input: data pairs
   $((\boldsymbol{X}_1, Y_1), (\boldsymbol{X}_2, Y_2), \cdots, (\boldsymbol{X}_t, Y_t))$
2: Construct $(\boldsymbol{V}_{t,X}, X_t)$ for each $X$, where $\boldsymbol{V}_{t,X}$ is the value of parent of $X$ at round $t$, $X_t$ is the value of $X$ at round t.
3: **for** $X \in \boldsymbol{X} \cup \{Y\}$ **do**
4:    $M_{t,X} = M_{t-1,X} + \boldsymbol{V}_{t,X}\boldsymbol{V}_{t,X}^\top$, calculate $\hat{\boldsymbol{\theta}}_{t,X}$ by solving $\sum_{i=1}^t (X_i - f_X(\boldsymbol{V}_{i,X}^T \hat{\boldsymbol{\theta}}_{t,X}))\boldsymbol{V}_{i,X} = 0$.
5: **end for**
6: **return** $\hat{\boldsymbol{\theta}}_t$.

---

For CCPE-BGLM, we define its $q_a^{(L)}$ as follows. Let $D = \max_{X \in \boldsymbol{X} \cup \{Y\}} |\boldsymbol{Pa}(X)|$. For node $\boldsymbol{S} \subseteq \boldsymbol{X}$, let $\ell_{\boldsymbol{S}} = |N_{\boldsymbol{S},Y}|$. Then for $a = do(\boldsymbol{S} = \boldsymbol{s})$, we define

$$q_a^{(L)} = \frac{1}{\ell_{\boldsymbol{S}}^2 D^3}. \tag{7}$$

Intuitively, based on Lemma 2 and $\ell_{\boldsymbol{S}} = |N_{\boldsymbol{S},Y}|$, a large $\ell_{\boldsymbol{S}}$ means that the right-hand side of Inequality (6) could be large, and thus it is difficult to estimate $\mu_a$ accurately. Hence the term $q_a^{(L)}$ represents how easy it is to estimate for action $a$. Note that $q_a^{(L)}$ only depends on the graph structure and set $\boldsymbol{S}$. We can define $m^{(L)}$ and $m_{\varepsilon,\Delta}^{(L)}$ with respect to $q_a^{(L)}$'s by Definitions 1 and 2. We use two confidence radius terms as follows, one from the estimate of the observational data, and the other from the estimate of the interventional data.

$$\beta_O^a(t) = \frac{\alpha_O M^{(1)} D^{1.5}}{\kappa\sqrt{\eta}}\sqrt{\frac{1}{q_a^{(L)}t} \log \frac{3nt^2}{\delta}}, \beta_I(t) = \alpha_I\sqrt{\frac{1}{t} \log \frac{|\boldsymbol{A}| \log(2t)}{\delta}}. \tag{8}$$

Parameters $\alpha_O$ and $\alpha_I$ are exploration parameters for our algorithm. For a theoretical guarantee, we choose $\alpha_O = 6\sqrt{2}$ and $\alpha_I = 2$, but more aggressive $\alpha_O$ and $\alpha_I$ could be used in experiments. (e.g. Mason et al. (2020), Kaufmann et al. (2016), Jamieson et al. (2013)) The sample complexity of CCPE-BGLM is summarized in the following theorem.

**Theorem 1.** *With probability* $1 - \delta$, *our* CCPE-BGLM$(G, \boldsymbol{A}, \varepsilon, \delta/2)$ *returns an* $\varepsilon$-*optimal arm with sample complexity*

$$T = O\left( H_{m_{\varepsilon,\Delta}^{(L)}} \log \frac{|\boldsymbol{A}|H_{m_{\varepsilon,\Delta}^{(L)}}}{\delta} \right), \tag{9}$$

*where* $m_{\varepsilon,\Delta}^{(L)}, H_{m_{\varepsilon,\Delta}^{(L)}}$ *are defined in Definition 2 and Eq.(3) in terms of* $q_a^{(L)}$'s *for* $a \in \boldsymbol{A} \setminus \{do()\}$ *defined in Eq.(7).*

If we treat the problem as a naive $|\boldsymbol{A}|$-arms bandit, the sample complexity of LUCB1 is $\widetilde{O}(H) = \widetilde{O}(\sum_{a \in \boldsymbol{A}} \frac{1}{\max\{\Delta_a, \varepsilon/2\}^2})$, which may contain an exponential number of terms. Now note that $q_a^{(L)} \geq \frac{1}{n^5}$, it is easy to show that $m_{\varepsilon,\Delta}^{(L)} \leq 2n^5$. Hence $H_{m_{\varepsilon,\Delta}^{(L)}}$ contains only a polynomial number of terms. Other causal bandit algorithms also suffer an exponential term, unless they rely on a strong and unreasonable assumption as describe in the related work. We achieve an exponential speedup by (a) a specifically designed algorithm for the BGLM model, and (b) interleaving observation and intervention and making the algorithm fully adaptive.

The idea of the analysis is as follows. First, for the $m_{\varepsilon,\Delta}$ hard actions, we rely on the adaptive LUCB to identify the best, and its sample complexity according to LUCB is $O(H_{m_{\varepsilon,\Delta}^{(L)}} \log(|\boldsymbol{A}|H_{m_{\varepsilon,\Delta}^{(L)}}/\delta))$.

Next, for easy actions, we rely on the observational data to provide accurate estimates. According to Eq.(4), every easy action $a$ has the property that $q_a \cdot \max\{\Delta_a, \varepsilon/2\}^2 \geq 1/H_{m_{\varepsilon,\Delta}}$. Using this property together with Lemma 2, we would show that the sample complexity for estimating easy action rewards is also $O(H_{m_{\varepsilon,\Delta}^{(L)}} \log(|\boldsymbol{A}|H_{m_{\varepsilon,\Delta}^{(L)}}/\delta))$. Finally, the interleaving of observations and interventions keep the samply complexity in the same order.

## 5 COMBINATORIAL PURE EXPLORATION FOR GENERAL GRAPHS

### 5.1 CPE ALGORITHM FOR GENERAL GRAPHS

In this section, we apply a similar idea to the general graph setting, which further allows the existence of hidden variables. The first issue is how to estimate the causal effect (or the do effect) $\mathbb{E}[Y \mid do(\boldsymbol{S} = \boldsymbol{s})]$ in general causal graphs from the observational data. The general concept of identifiability (Pearl, 2009) is difficult for sample complexity analysis. Here we use the concept of *admissible sequence* (Pearl, 2009) to achieve this estimation.

**Definition 3** (Admissible sequence). *An admissible sequence for general graph $G$ with respect to $Y$ and $\boldsymbol{S} = \{X_1, \cdots, X_k\} \subseteq \boldsymbol{X}$ is a sequence of sets of variables $\boldsymbol{Z}_1, \cdots \boldsymbol{Z}_k \subseteq \boldsymbol{X}$ such that*

*(1) $\boldsymbol{Z}_i$ consists of nondescendants of $\{X_i, X_{i+1}, \cdots, X_k\}$,*

*(2) $(Y \perp\!\!\!\perp X_i \mid X_1, \cdots, X_{i-1}, \boldsymbol{Z}_1, \cdots, \boldsymbol{Z}_i)_{G_{\underline{X}_i, \overline{X}_{i+1}, \cdots, \overline{X}_k}}$ , where $G_{\underline{X}}$ means graph $G$ without out-edges of $X$, and $G_{\overline{X}}$ means graph $G$ without in-edges of $X$.*

Then, for $\boldsymbol{S} = \{X_1, \cdots, X_k\}$, $\boldsymbol{s} = \{x_1, \cdots, x_k\}$, we can calculate $\mathbb{E}[Y \mid do(\boldsymbol{S} = \boldsymbol{s})]$ by

$$\mathbb{E}[Y \mid do(\boldsymbol{S} = \boldsymbol{s})] = \sum_{\boldsymbol{z}} P(Y = 1 \mid \boldsymbol{S} = \boldsymbol{s}, \boldsymbol{Z}_i = \boldsymbol{z}_i, i \leq k)$$
$$\cdot P(\boldsymbol{Z}_1 = \boldsymbol{z}_1) \cdots P(\boldsymbol{Z}_k = \boldsymbol{z}_k \mid \boldsymbol{Z}_i = \boldsymbol{z}_i, X_i = x_i, i \leq k-1), \tag{10}$$

where $\boldsymbol{z}$ means the value of $\cup_{i=1}^k \boldsymbol{Z}_i$, and $\boldsymbol{z}_i$ means the projection of $\boldsymbol{z}$ on $\boldsymbol{Z}_i$. For $a = do(\boldsymbol{S} = \boldsymbol{s})$ with $|\boldsymbol{S}| = k$, we use $\{\boldsymbol{Z}_{a,i}\}_{i=1}^k$ to denote the admissible sequence with respect to $Y$ and $\boldsymbol{S}$ , and $\boldsymbol{Z}_a = \cup_{i=1}^k \boldsymbol{Z}_{a,i}$. $Z_a = |\boldsymbol{Z}_a|$ and $Z = \max_a Z_a$. In this paper, we simplify $\boldsymbol{Z}_{a,i}$ to $\boldsymbol{Z}_i$ if there is no ambiguity.

For any $\boldsymbol{P} \subseteq \boldsymbol{X}$, denote $\boldsymbol{P}_t = \boldsymbol{X}_t|_{\boldsymbol{P}}$ as the projection of $\boldsymbol{X}_t$ on $\boldsymbol{P}$. We define

---

**Algorithm 3** CCPE-General($G, \boldsymbol{A}, \varepsilon, \delta$)

---

1: Input:causal graph $G$, action set $\boldsymbol{A}$, parameter $\varepsilon, \delta$, admissible sequence $\{(Z_a)_i\}$ for each action $a \in \boldsymbol{A}$

2: Initialize $t = 1, T_a = 0, T_{a,\boldsymbol{z}} = 0, N_a = 0, \hat{\mu}_a = 0$ for all arms $a \in \boldsymbol{A}, \boldsymbol{z} \in \{0,1\}^z, z \in [|X|]$.

3: **for** $t = 1, 2, \cdots,$ **do**

4:    $a_h^{t-1} = \mathrm{argmax}_{a \in \boldsymbol{A}} \hat{\mu}_a^{t-1}, a_l^{t-1} = \mathrm{argmax}_{a \in \boldsymbol{A} \backslash a_h^{t-1}}(U_a^{t-1})$

5:    **if** $U_{a_l^{t-1}} \leq L_{a_h^{t-1}} + \varepsilon$ **then**

6:       **return** $a_h^{t-1}$

7:    **end if**

8:    /* Step 1. Conduct a passive observation and estimate from the observational data */

9:    Perform $do()$ operation and observe $\boldsymbol{X}_t$ and $Y_t$. For $a = do(), N_a = N_a + 1$.

10:    **for** $a = do(\boldsymbol{S} = \boldsymbol{s}) \in \boldsymbol{A} \setminus \{do()\}$ with an admissible sequence and $\boldsymbol{S} = \{X_1, \cdots, X_k\}, \boldsymbol{s} = \{x_1, \cdots, x_k\}$ **do**

11:       Estimate $\hat{\mu}_{O,a}$ using (14) and $[L_{O,a}^t, U_{O,a}^t] = [\hat{\mu}_{O,a} - \beta_O^a(T_a, t), \hat{\mu}_{O,a} + \beta_O^a(T_a)]$. /* $\beta_O^a(t)$ is defined in Eq.(16), $T_{a,\boldsymbol{z}}$ is defined in Eq.(11) and $T_a = \min_{\boldsymbol{z}} T_{a,\boldsymbol{z}}$. */

12:    **end for**

13:    /* Step 2. Do two interventions and estimate from the interventional data */

14:    Perform actions $a_l^{t-1}$ and $a_h^{t-1}$, get the reward $Y_t^{(l)}$ and $Y_t^{(h)}$.

15:    $N_{a_l^{t-1}} = N_{a_l^{t-1}} + 1, N_{a_h^{t-1}} = N_{a_h^{t-1}} + 1$.

16:    For $a \in \{a_l^{t-1}, a_h^{t-1}, do()\}$, update the empirical mean $\hat{\mu}_{I,a}$ as Line 16 in Algorithm 1.

17:    Update $[L_{I,a}^t, U_{I,a}^t] = [\hat{\mu}_{I,a} - \beta_I(N_a), \hat{\mu}_{I,a} + \beta_I(N_a)]$. /* $\beta_I(t)$ is defined in Eq.(16) */

18:    /* Step 3. Merge the observational estimate and the interventional estimate */

19:    For $a \in \boldsymbol{A}$, calculate $[L_a^t, U_a^t] = [L_{O,a}^t, U_{O,a}^t] \cap [L_{I,a}^t, U_{I,a}^t]$ and $\hat{\mu}_a^t = \frac{L_a^t + U_a^t}{2}$.

20: **end for**

---

$$T_{a,\boldsymbol{z}} = \sum_{j=1}^t \mathbb{I}\{\boldsymbol{S}_j = \boldsymbol{s}, (\boldsymbol{Z}_a)_j = \boldsymbol{z}\}, r_{a,\boldsymbol{z}}(t) = \frac{1}{T_{a,\boldsymbol{z}}} \sum_{j=1}^t \mathbb{I}\{\boldsymbol{S}_j = \boldsymbol{s}, (\boldsymbol{Z}_a)_j = \boldsymbol{z}\} Y_j \tag{11}$$

$$n_{a,\boldsymbol{z},l}(t) = \sum_{j=1}^t \mathbb{I}\{(\boldsymbol{Z}_i)_j = \boldsymbol{z}_i, (X_i)_j = x_i, i \leq l-1\} \tag{12}$$

$$p_{a,\boldsymbol{z},l}(t) = \frac{1}{n_{a,\boldsymbol{z},l}(t)} \sum_{j=1}^t \mathbb{I}\{(\boldsymbol{Z}_l)_j = \boldsymbol{z}_l, (\boldsymbol{Z}_i)_j = \boldsymbol{z}_i, (X_i)_j = x_i, i \leq l-1\} \tag{13}$$

where the $r_{a,\boldsymbol{z}}(t)$ and $p_{a,\boldsymbol{z},l}(t)$ are the empirical mean of $P(Y \mid \boldsymbol{S} = \boldsymbol{s}, \boldsymbol{Z}_a = \boldsymbol{z})$ and $P(\boldsymbol{Z}_l = \boldsymbol{z}_l \mid \boldsymbol{Z}_i = \boldsymbol{z}_i, X_i = x_i, i \leq l-1)$. Also, we denote $T_a = \min_{\boldsymbol{z}} T_{a,\boldsymbol{z}}$. Using the above Eq.(10), we estimate each term of the right-hand side for every $\boldsymbol{z} \in \{0,1\}^{Z_a}$ to obtain an estimate for $\mathbb{E}[Y \mid a]$ as follows:

$$\hat{\mu}_{O,a} = \sum_{\boldsymbol{z}} r_{a,\boldsymbol{z}}(t) \prod_{l=1}^k p_{a,\boldsymbol{z},l}(t). \tag{14}$$

For general graphs, there is no efficient algorithm to determine the existence of the admissible sequence and extract it when it exists. But we could rely on several methods to find admissible sequences in some special cases. First, we can find the *adjustment set*, a special case of admissible sequences. For a causal graph $G$, $\boldsymbol{Z}$ is an adjustment for variable $Y$ and set $\boldsymbol{S}$ if and only if $P(Y = 1 \mid do(\boldsymbol{S} = \boldsymbol{s})) = \sum_{\boldsymbol{z}} P(Y = 1 \mid \boldsymbol{S} = \boldsymbol{s}, \boldsymbol{Z} = \boldsymbol{z})P(\boldsymbol{Z} = \boldsymbol{z})$. There is an efficient algorithm for deciding the existence of a minimal adjustment set with respect to any set $\boldsymbol{S}$ and $Y$ and finding it (van der Zander et al., 2019). Second, for general graphs without hidden variables, the admissible sequence can be easily found by $\boldsymbol{Z}_j = \boldsymbol{Pa}(X_j) \setminus (\boldsymbol{Z}_1 \cup \cdots \boldsymbol{Z}_{j-1} \cup X_1 \cdots \cup X_{j-1})$ (Theorem 4 in the Appendix). Finally, when the causal graph satisfies certain properties, there exist algorithms to decide and construct admissible sequences Dawid & Didelez (2010).

Algorithm 3 provides the pseudocode of our algorithm CCPE-General, which has the same framework as Algorithm 1. The main difference is in the first step of updating observational estimates, in which we rely on the do-calculus formula Eq.(10).

For an action $a = do(\boldsymbol{S} = \boldsymbol{s})$ without an admissible sequence, define $q_a^{(G)} = 0$, meaning that it is hard to be estimated through observation. Otherwise, define $q_a$ as:

$$q_a^{(G)} = \min_{\boldsymbol{z}}\{q_{a,\boldsymbol{z}}\}, \text{where } q_{a,\boldsymbol{z}} = P(\boldsymbol{S} = \boldsymbol{s}, \boldsymbol{Z}_a = \boldsymbol{z}), \forall \boldsymbol{z} \in \{0,1\}^{Z_a}. \quad (15)$$

Similar to CCPE-BGLM, for $a = do(\boldsymbol{S} = \boldsymbol{s})$ with $|\boldsymbol{S}| = k$, we use observational and interventional confidence radius as:

$$\beta_O^a(n,t) = \alpha_O \sqrt{\frac{1}{n} \log \frac{20k|\boldsymbol{A}|Z_a I_a \log(2t)}{\delta}}; \beta_I(t) = \alpha_I \sqrt{\frac{1}{n} \log \frac{|\boldsymbol{A}| \log(2t)}{\delta}}, \quad (16)$$

where $\alpha_O$ and $\alpha_I$ are exploration parameters, and $I_a = 2^{Z_a}$. For a theoretical guarantee, we will choose $\alpha_O = 8$ and $\alpha_I = 2$. Our sample complexity result is given below.

**Theorem 2.** *With probability $1 - \delta$, CCPE-General$(G, \boldsymbol{A}, \varepsilon, \delta/5)$ returns an $\varepsilon$-optimal arm with sample complexity*

$$T = O\left(H_{m_{\varepsilon,\Delta}^{(G)}} \log \frac{|\boldsymbol{A}| H_{m_{\varepsilon,\Delta}^{(G)}}}{\delta}\right), \quad (17)$$

*where $m_{\varepsilon,\Delta}^{(G)}, H_{m_{\varepsilon,\Delta}^{(G)}}$ are defined in Definitions 2 and 3 in terms of $q_a^{(G)}$'s defined in Eq.(15).*

Comparing to LUCB1, since $m_{\varepsilon,\Delta}^{(G)} \leq |\boldsymbol{A}|$, our algorithm is always as good as LUCB1. It is easy to construct cases where our algorithm would perform significantly better than LUCB1. Comparing to other causal bandit algorithms, our algorithm also performs significantly better, especially when $m_{\varepsilon,\Delta}^{(G)} \ll m^{(G)}$ or the gap $\Delta_a$ is large relative to $\varepsilon$. Some causal graphs with candidate action sets and valid admissible sequence are provided in the Appendix A, and more discussion is in the Appendix.

## 5.2 LOWER BOUND FOR THE GENERAL GRAPH CASE

To show that our CCPE-General algorithm is nearly minimax optimal, we provide the following lower bound, which is based on parallel graphs. We consider the following class of parallel bandit instance $\xi$ with causal graph $G = (\{X_1, \cdots, X_n, Y\}, E)$: the action set is $\boldsymbol{A} = \{do(X_i = x) \mid x \in \{0,1\}, 1 \leq i \leq n\} \cup \{do()\}$. The $q_a^{(G)}$ in this case is reduced to $q_{do(X_i=x)}^{(G)} = P(X_i = x)$ and $q_{do()} = 1$. Sort the action set as $q_{a_1}^{(G)} \cdot \max\{\Delta_{a_1}, \varepsilon/2\}^2 \leq q_{a_2}^{(G)} \cdot \max\{\Delta_{a_2}, \varepsilon/2\}^2 \leq \cdots \leq q_{a_{2n+1}}^{(G)} \cdot \max\{\Delta_{a_{2n+1}}, \varepsilon/2\}^2$. Let $p_{\min} = \min_{\boldsymbol{x} \in \{0,1\}^n} P(Y = 1 \mid \boldsymbol{X} = \boldsymbol{x}), p_{\max} = \max_{\boldsymbol{x} \in \{0,1\}^n} P(Y = 1 \mid \boldsymbol{X} = \boldsymbol{x})$. Let $p_{\max} + 2\Delta_{2n+1} + 2\varepsilon \leq 0.9, p_{\min} + \Delta_{\min} \geq 0.1$.

**Theorem 3.** *For the parallel bandit instance class $\xi$ defined above, any $(\varepsilon, \delta)$-PAC algorithm has expected sample complexity at least*

$$\Omega\left(\left(H_{m_{\varepsilon,\Delta}^{(G)}-1} - \frac{1}{\min_{i < m_{\varepsilon,\Delta}^{(G)}} \max\{\Delta_{a_i}, \varepsilon/2\}^2} - \frac{1}{\max\{\Delta_{do()}, \varepsilon/2\}^2}\right) \log\left(\frac{1}{\delta}\right)\right). \quad (18)$$

Theorem 3 is the first gap-dependent lower bound for causal bandits, which needs brand-new construction and technique. Comparing to the upper bound in Theorem 2, the main factor $H_{m_{\varepsilon,\Delta}^{(G)}}$ is the same, except that the lower bound subtracts several additive terms. The first term $H_{m_{\varepsilon,\Delta}-1}$ is almost equal to $H_{m_{\varepsilon,\Delta}}$ appearing in Eq.(17), except the it omits the last and the smallest additive term in $H_{m_{\varepsilon,\Delta}}$. The second term is to eliminate one term with minimal $\Delta_{a_i}$, which is common in multi-armed bandit. (Lattimore (2018),Karnin et al. (2013a)) The last term is because $do()$'s reward must be in-between $\mu_{do(X_i=0)}$ and $\mu_{do(X_i=1)}$ and thus cannot be the optimal arm.

## 6 FUTURE WORK

There are many interesting directions worth exploring in the future. First, how to improve the computational complexity for CPE of causal bandits is an important direction. Second, one can consider developing efficient pure exploration algorithms for causal graphs with partially unknown graph structures. Lastly, identifying the best intervention may be connected with the markov decision process and studying their interactions is also an interesting direction.

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
