# OpenReview forum: "Combinatorial Pure Exploration of Causal Bandits"
_ICLR.cc/2023/Conference — ICLR 2023 poster_

### Official Review · Reviewer_jo8z · 2022-10-24

**Confidence:** 2
**Correctness:** 4
**Technical Novelty And Significance:** 4
**Empirical Novelty And Significance:** 3
**Recommendation:** 8

**Clarity, Quality, Novelty And Reproducibility:**

Quality and novelty: This paper resolves the combinatorial pure exploration in casual bandits. It proposes novel algorithms for BGLM and generalized graphs and provides gap-dependent upper bounds on the sample complexity, which are accompanied by gap-dependent lower bounds on sample complexity for parallel graphs. Overall, this is a solid piece of work.

Clarity: Long pseudocodes in the main text do not provide intuition about what is going on. It might be better to move the pseudocode of Algorithm 3 to the appendix and use the opened-up space for numerical results or additional insights.

Why does Algorithm 3 require a casual parallel graph G as input (line 1)?


**Strength And Weaknesses:**

-Analyses pure exploration of combinatorial casual bandits in a general setting. Extends the results of prior works to general graphs with the possibility of unobserved variables.

-Provides gap-dependent bounds on the sample complexity for the binary generalized linear model (BGLM). Unlike prior works, whose complexity grows exponentially, for BGLM, their sample complexity scales only polynomially in the graph size. Full adaptivity of the algorithm by interleaving exploration and exploitation is a key contributor to this improvement.

-Provides an analysis of the general case where hidden variables exist.

-Shows a lower bound for parallel graphs, establishing the near minimax optimality of the combinatorial casual pure exploration algorithm.


**Summary Of The Paper:**

-Proposes gap-dependent bounds on adaptive pure exploration algorithms.

-Algorithms do not require prior knowledge of causal inference distributions.

-Allows combinatorial interventions.

-Applicable to causal graphs with hidden variables.

-Studies sample complexity in both fixed confidence and fixed budget settings.

-Proposes the gap-dependent observation threshold as an intuitive measure of problem difficulty.

-Proposes adaptive algorithms that interleave observation and intervention for efficient sample learning of optimal interventions.


**Summary Of The Review:**

A good paper that addresses several important issues in the combinatorial pure exploration of casual bandits with solid theoretical results.

---

> ### Author Response · Authors · 2022-11-15
> **Response to Reviewer jo8z**
>
> Thanks for your constructive comment, which we will incorporate into our revised paper. Please see our response below.
>
> 1."Long pseudocodes in the main text do not provide intuition about what is going on."
>
> We will add more intuition and explanations about Alg. 3 in our final version, possibly moving the pseudocode into the appendix if necessary.
>
>
> 2."Why does Algorithm 3 require a casual parallel graph G as input (line 1)?"
>
> It is a typo. It should just be "causal graph $G$". We have fixed it in our revised version.

---

### Official Review · Reviewer_okVv · 2022-10-25

**Confidence:** 4
**Correctness:** 3
**Technical Novelty And Significance:** 3
**Empirical Novelty And Significance:** Not applicable
**Recommendation:** 8

**Clarity, Quality, Novelty And Reproducibility:**

The paper heavily relies on Feng & Chen (2022) for some reason, even pulling lemmas from that unpublished manuscript without proofs.

Compared to Lattimore 2016, I think the contribution is novel.

**Strength And Weaknesses:**

They propose a solution to a very special case of a very hard problem. I think the results are still interesting.

The linear setting is a bit restrictive especially when combined with the assumption that all parameters are nonnegative (can this assumption be removed?)

The general graph case is a bit rushed. For example, how often do we have that all candidate actions have valid admissable sets? What if only some of them do? The algorithm should have been robust to this. Is there a way to incorporate this? Also it would be good to draw some graphs where the assumptions hold.

**Summary Of The Paper:**

The authors extend Lattimore 2016 to obtain a gap-dependent bound (not regret since they operate in the pure exploration setting) bandit for parallel bandit graph, and also extend it to arbitrary graphs, so long as there is an admissable set. For example, if there is a valid back-door adjustment for the actions.

**Summary Of The Review:**

Thank you for your submission. I did enjoy reading it. I have a few comments/questions below.

Why are the weights in the network always non-negative? Isn't it to very strong assumption that the algorithm always knows that setting values to 1s are the only possible optimal interventions?

The paper borrows several assumption and even lemmas from Feng & Chen (2022). But this is an unpublished manuscript on arxiv. Specifically for the borrowed lemmas, it would be good to have proofs of them self contained in this submission. I couldn't find these proofs are given in the Appendix either.

Some intuition about the definition of Gap-dependent observation thershold would be nice to have, similar to the baseline m value borrowed from Lattimore.

Please add some causal graphs with candidate action sets for which admissable sequences or adjustment sets exist with latent confounders.

---

> ### Author Response · Authors · 2022-11-15
> **Response to Reviewer okVv**
>
> Thanks for your constructive comment, which we will incorporate into our revised paper. Please see our response below.
>
> 1."The linear setting is a bit restrictive especially when combined with the assumption that all parameters are nonnegative."
> As we answered to Reviewer SLrk, we can remove the assumptions that values in $\theta_X$ are nonnegative and can allow both positive and negative values,
> 	as long as the probability $P(X=1 | Pa(X)=pa(X))$ it determined by $f_X$ and $e_X$ is a valid probability value.
>
> 2."The algorithm should have been robust to the situation that not all actions have a valid admissible sequence."
>
> Thanks for pointing this out. Indeed, we do not need all candidate actions to have admissible sequences.
> In our algorithm for general graphs, for actions with valid admissible sequences, we use observations to update their estimations.
> For actions without valid admissible sequences, we simply use their intervention results since it is difficult to estimate their effect by the observational data.
> We will clarify this point in the revised version.
>
> 3."It would be good to have proofs for preprint."
>
> We provide the proofs of Lemma 2 and 3 in Appendix D.3 and D.4 in new version.
>
> 4. "Some intuition about the definition of Gap-dependent observation threshold would be nice to have"
>
> Intuitively, when considering the gap, $q_a\cdot \max(\Delta_a,\varepsilon/2)^2$ represents how easily an action $a$ could  be to be distinguished from the optimal arm.
> Thus in $m_{\varepsilon,\Delta}$ and $H_\tau$ defined in the paper, we sort the action by $q_a\cdot \max(\Delta_a,\varepsilon/2)^2$, and regard actions with large $q_a\cdot \max(\Delta_a,\varepsilon/2)^2$ as "easy" actions.
> When not considering the gap, $q_a$ only measures how easily we can obtain observational data for action $a$ but does not directly relating $a$ with the optimal action
> 	using the gap.
> Technically, this is equivalently as treating the gap $\Delta_a$ as $0$.
> In this case, the gap-dependent observation threshold $m_{\varepsilon,\Delta}$ degenerates into the standard observation threshold $m$.
> We will add this discussion in our main text in the revised version.
>
> 5."Please add some causal graphs with candidate action sets for which admissable sequences or adjustment sets exist with latent confounders."
>
> We add some graphs in Appendix A. Moreover, as pointed out in item 2 above, we do not need all actions to have valid admissible sequences or adjustment sets.

---

### Official Review · Reviewer_SLrk · 2022-10-28

**Confidence:** 3
**Correctness:** 4
**Technical Novelty And Significance:** 2
**Empirical Novelty And Significance:** Not applicable
**Recommendation:** 6

**Clarity, Quality, Novelty And Reproducibility:**

Quality: Overall the paper is good.

Clarity: Generally speaking, I found the writing and presentation clear.

Originality: The authors employ strategies (LUCB alg.) and results (BLGM concentrations) from past works, but in a non-trivial manner.

Reproducibility: The authors provided proofs in the supplementary material, and proof sketches in the main paper.  I did not check the supplementary material.  I did not spot any issues with correctness.


**Strength And Weaknesses:**

### Major strengths

1. The authors showed that for a particular parametric class of models (BGLM with bounded derivatives), since the difficulty of estimating the conditional mean of Y can be characterized by graph properties in such a way that actions can be separated into “easy” and “hard” actions, an LUCB style algorithm can be employed that significantly improves the sample complexity.

2. The authors also generalize that strategy for the more general case with hidden nodes and for which the conditional distributions are not parameterized, and show the sample complexity of that algorithm is nearly minimax optimal.

### Minor strengths

1. The authors propose a fixed-budget pure exploration algorithm and analyze its regret (in supplementary material).

2. The authors include some toy experiments for their methods (in supplementary material).

### Major weaknesses
1. The storage and per-round computational complexities can be exponential in $n$ even for parameterized setting with the BGLM model.  Separate statistics are maintained for each action (eg Alg. 1 line 2) and updated (eg Alg. 1 line 11).  The action set is allowed to be an arbitrary subset of the powerset of $X$.

### Minor weaknesses
1. The assumptions for the BGLM model used are quite strong – $\theta$ parameters are non-negative, $f_X$ monotone, so that for any subset of nodes, it is immediate that the best intervention is to set all variables in that subset to value of 1.  Even for the slightly more general BGLM model without the same restrictions on \theta or $f_X$ (but with bounded derivatives), it seems non-trivial to modify Algorithm 1 short of treating each realization as a separate action, growing the action set (and consequently storage complexity and per-round computational complexities) exponentially.

2. The readability could be improved by modifying notation or adding more verbal descriptors reminding the reader what notations refer to, especially those used sparingly -- eg $P_{S,Y}$ used to specify nodes along paths but P already used for probabilities; \sigma introduced on page 5 for expected values and used once in Lemma 2 but already have \mu notation for that.  The pseudocode of Alg 1 and 3 are heavy with notation and equations – the precise formulas could be discussed in the main text and referenced (or simplified).

3. The storage and per-round computational complexities of the methods were not formally discussed in the main sections.



### Questions
1. Pg 3 Can you clarify what the sentence “To represent the native probability, …” means

2. Pg 3 Can you clarify why in the BLGM model the probability mass functions are random?  The X’s would be random even without the $e_X$ term.  More concerning to me is that you have an additive random term, but you are assuming that the sum is constrained in [0,1] to be a valid probability mass.  To me at least, that seems to induce a strange constraint of f_X and $e_X$ since if the terms are independent, then both need to be bounded.  If $e_X$ is bounded, then it is sub-Gaussian.

3. Pg 4 “We assume that Xi’s are Bernoulli random variables when we do not intervene any variables” – What does that assumption mean?  They would have marginal Bernoulli distributions since they are binary valued, but they would have joint dependence.



**Summary Of The Paper:**

The authors study the problem of causal bandits with (arbitrary) combinatorial action spaces in both fixed-confidence and fixed-budget pure exploration (simple regret) settings.  The causal graph is assumed to be known and all variables are binary valued.  The authors consider two cases.  For the first, the causal model is parameterized, with which parameters for most actions can be estimated from observational data and the rest are estimated using interventions.  The algorithm is based on the LUCB algorithm.  For the second (more general) case, the causal model is not parameterized and hidden nodes are available; the proposed algorithm overall has a similar structure as in the first case.   The authors also show that the sample complexity of the latter algorithm is nearly minimax optimal.

**Summary Of The Review:**

Overall I think the submission is good.  The topic, at the intersection of causal inference and multi-armed bandits can be of interest to multiple communities.  The authors adapt the LUCB algorithm in an interesting way for the causal bandit problem.  I have concerns that the storage and per-round computational complexity of the algorithm may render the algorithm primarily of theoretical interest, but nonetheless think the submission is good.

---

> ### Author Response · Authors · 2022-11-15
> **Response to Reviewer SLrk**
>
> Thanks for your constructive comment, which we will incorporate into our revised paper. Please see our response below.
>
> 1.The assumption on the non-negative values for parameters $\theta_X$ in the BGLM model is quite strong.
> Thanks for your question. We notice that our results on BGLM can hold with negative weights $\theta_X$, as long as $f_X(\theta_X^T Pa(X))+e_X$ is a valid probability. We need $f_X$ to be strictly increasing to make our MLE estimation correct, which is also needed in generalized linear bandit [1,2]. We will clarify this point in our revised version.
>
> 2."The storage and per-round computational complexities can be exponential in $n$ ." and "The storage and per-round computational complexities of the methods were not formally discussed in the main sections"
> This is a good and important question. The causal bandit does not have the nice structures like "base arms" in the classical combinatorial semi-bandit, so it is much harder to achieve polynomial sample complexity and polynomial computational/storage complexity at the same time. Some previous studies on causal bandits [3,4] also suffer from this problem. How to address this is an interesting direction for future research. In practice, we can collect a batch of data and update the estimation of observation per batch, and this may alleviate the computational complexity significantly without compromising the correctness of the algorithm. We will add some discussions in our final version.
>
> 3."The readability could be improved"
> We will try to simplify or modify some notations to make paper clear.
>
> Also, definition $\sigma(\hat{\theta}^{t}, a)=\hat{\mu}^t_{O,a}$
> is to show that the estimation $\hat{\mu}_{O,a}^t$   at round t is calculated by parameters $\hat{\theta^t}$.
>
> Notation $\hat{\mu}_{O,a}^t$ does not show its dependency on $\hat{\theta^t}$, and we need a general function form of $\sigma(\theta, a)$ because of Lemma 2, which states the effect of changing parameters $\theta$ on the reward. We will clarify this point, and also check and make sure our notations throughout the paper are easy to understand.
>
> 4. Other questions.
>
> a)What does "To represent the native probability" mean?
> In causal bandits, nodes will have some randomness not caused by the parents. We use a global variable $X_1$ to represent this effect.
>
> b) "Why the probability mass functions are random in BGLM model?"
> We inherit this error term from the classical linear models or generalized linear models. We feel that we could handle this error term in our BGLM and thus leave this error term there. To make it as a valid probability, we indeed need the $f_X$ and $e_X$ are bounded and thus sub-Gaussian.
>
> c)What does "We assume that Xi’s are Bernoulli random variables when we do not intervene any variables." mean?
> We should make it more clear. As the reviewer pointed out, it is random variable with marginal Bernoulli distribution. All variables $X_i$'s have joint
> distributions and they may not be independent.
>
>
> [1] Li, L. et al. Provably optimal algorithms
> for generalized linear contextual bandits.  (ICML 2017)
>
> [2] Filippi et al. Parametric Bandits: The Generalized Linear Case (Neurips 2010)
>
> [3]Vineet Nair, Vishakha Patil, and Gaurav Sinha. Budgeted and non-budgeted causal bandits. (AISTATS 2021)
>
> [4]Yangyi Lu et al. Regret analysis of bandit problems with causal background knowledge. (UAI 2020)

---

### Decision · Program_Chairs · 2023-01-20

**Decision:**

Accept: poster

**Justification For Why Not Higher Score:**

I think there are still too many restrictive assumptions and this is not a paper that solves a broadly interesting open problem.

**Justification For Why Not Lower Score:**

The results are interesting enough to merit publication/presentation.

**Metareview: Summary, Strengths And Weaknesses:**

The authors extend Lattimore 2016 to to a larger classed of graphs (the graph could be arbitrary as long as there is an admissible set) for the setting of combinatorial action spaces in both fixed-confidence and fixed-budget pure exploration (simple regret) scenarios.

Although some of the reviewers were concerned about limitations in the model (say linearity of positivity fo the parameters), they found the results and the presentation overall convincing.


**Note From Pc:**

if the above contains the word "oral" or "spotlight" please see: "oral" presentation means -> notable-top-5% and "spotlight" means -> notable-top-25%. As stated in our emails, we are disassociating presentation type from AC recommendations